# Coordinated microRNA/mRNA Expression Profiles Reveal Unique Skin Color Regulatory Mechanisms in Chinese Giant Salamander (*Andrias davidianus*)

**DOI:** 10.3390/ani13071181

**Published:** 2023-03-28

**Authors:** Yanjie Guo, Weini Wu, Xueyi Yang

**Affiliations:** Life Science College, Luoyang Normal University, Luoyang 471934, China

**Keywords:** *Andrias davidianus*, albino individual, melanogenesis, pigmentation, *MITF* splicing, *SLC24A5* mutation

## Abstract

**Simple Summary:**

The molecular mechanisms of skin pigmentation in *Andrias davidianus* are not clear. In this study, we identified two albino individuals and found that *Andrias davidianus* had distinct regulatory mechanisms of skin pigmentation that differed from other vertebrates. The key signaling pathway and transcription factors related to melanin synthesis in other vertebrates did not play a significant role in *Andrias davidianus*. *MITF* mRNA in *Andrias davidianus* had a unique splicing form that was not reported in other vertebrates and a unique mutation existed in the *SLC24A5* gene in albino *Andrias davidianus*. The results contributed to understanding the molecular mechanism of skin pigmentation in *Andrias davidianus* and accelerating the acquisition process of *Andrias davidianus* species with specific body colors by genetic means, which will help to enrich the aquaculture market of *Andrias davidianus*.

**Abstract:**

The Chinese giant salamander (*Andrias davidianus*) has been increasingly popular in the aquaculture market in China in recent years. In the breeding process of *Andrias davidianus*, we found that some albino individuals were extremely rare and could not be inherited stably, which severely limits their commercialization in the aquaculture market. In this study, we performed transcriptome and small RNA (sRNA) sequencing analyses in the skin samples of wild-type (WT) and albino (AL) *Andrias davidianus*. In total, among 5517 differentially expressed genes (DEGs), 2911 DEGs were down-regulated in AL, including almost all the key genes involved in melanin formation. A total of 25 miRNAs were differentially expressed in AL compared to WT, of which 17 were up-regulated. Through the integrated analysis, no intersection was found between the target genes of the differentially expressed miRNAs and the key genes for melanin formation. Gene Ontology (GO) and KEGG pathway analyses on DEGs showed that these genes involved multiple processes relevant to melanin synthesis and the key signal pathway MAPK. Interestingly, the transcription factors SOX10 and PAX3 and the Wnt signaling pathway that play a key role in other species were not included, while the other two transcription factors in the SOX family, SOX21 and SOX7, were included. After analyzing the key genes for melanin formation, it was interesting to note an alternative splicing form of the *MITF* in WT and a critical mutation of the *SLC24A5* gene in AL, which might be the main reason for the skin color change of *Andrias davidianus*. The results contributed to understanding the molecular mechanism of skin pigmentation in *Andrias davidianus* and accelerating the acquisition process of individuals with specific body colors by genetic means.

## 1. Introduction

The Chinese giant salamander (*Andrias davidianus*), which currently represents the largest amphibian worldwide, is of great scientific, ecological, and economic importance [1]. With the gradual sophistication of artificial propagation and breeding technology and the increasing enlargement of the breeding scale, the artificial breeding of *Andrias davidianus* in China has gradually emerged as a new variety with an extremely promising market. The *Andrias davidianus* mainly has brown or black skin, with very few individuals showing other colors. However, body color serves as an important economic indicator in aquaculture, and its diversity will certainly lead to significant economic benefits. We identified several albino individuals (complete or incomplete) during the breeding of *Andrias davidianus*, and giant salamanders with this special body color are particularly popular among consumers in the market as an ornamental animal. However, such albino individuals are quite scarce, and their body color cannot be stably inherited, which significantly limits their commercialization process in the aquaculture market.

The animal’s body color is an important phenotypic feature that is primarily determined by various pigments synthesized by chromatophores or pigment cells [2]. Up to now, six types of pigment cells have been reported in vertebrates that determine the animal’s body color by regulating its cell number and the production and release of pigments [3,4,5]. The skin color of *Andrias davidianus* (black or brown) is mainly determined by melanocytes. The synthesis of melanin mainly involves the tyrosine metabolic pathway, of which *MITF* is considered to be the master regulator that is able to phosphorylate the key enzymes involved in melanin synthesis, including tyrosinase (TYR), tyrosinase-related protein 1 (TYRP1), dopachrome tautomerase (DCT), and SILV (Silver Locus Protein Homolog), to activate their transcription [6,7]. The regulation of melanin synthesis is principally achieved through MITF. Transcription factors such as paired box 3 (PAX3), sex-determining region Y box 10 (SOX10), and lymphoid enhancer binding factor 1 (LEF1) can bind to the *MITF* upstream promoter to affect the formation of melanin by mediating *MITF* expression [8,9,10]. MAPK and Wnt signaling pathways are two major pathways involved in melanin formation that can affect melanin synthesis by regulating the activity or expression of *MITF* [11,12]. Other genes such as *OCA2*, *SLC24A5,* and *SLC45A2*-encoded ion-exchange proteins can influence melanin formation by regulating the concentrations of positive ions such as Na^+^, K^+^, and H^+^ in melanosomes [13,14,15]. Additionally, several noncoding RNAs, such as miR-206, miR-429, miR-137, and miR-330-5p, have been demonstrated to regulate *MITF* or to directly regulate key genes in the tyrosine metabolic pathway [16,17,18,19]. Conclusively, the melanin synthesis pathways have been extensively studied, and the results have revealed that these signaling pathways seem to be conserved in vertebrates [7,20,21]. However, the regulation of body color in *Andrias davidianus* has not yet been reported, and whether the regulation of body color in *Andrias davidianus* is specific or the same as in humans, mice, or other reported vertebrates remains unknown.

In the present study, through combined transcriptome and sRNA sequencing analyses, we found that the vast majority of key genes involved in melanin synthesis were differentially expressed between wild-type and albino *Andrias davidianus* with two different body colors, which is similar to the findings in other vertebrates. However, specificity was also identified in the regulation of body color in *Andrias davidianus*, mainly manifested as follows: (1) there were few differentially expressed miRNAs between the two types of *Andrias davidianus* with different body colors and almost no intersection between the target genes of those differentially expressed miRNAs and the key regulators of melanin synthesis, suggesting that miRNAs do not play a critical role in the body color regulation in *Andrias davidianus*; (2) by analyzing MAPK and Wnt signaling pathways, which are key pathways in other vertebrates, we found that the Wnt signaling pathway seems to have no function in body color regulation in *Andrias davidianus*; (3) PAX3 and SOX10, two key transcription factors for melanin synthesis, were not differentially expressed between the wild-type and albino *Andrias davidianus*, but we identified two other differentially expressed transcription factors of the SOX family (SOX7 and SOX21), showing consistency in their expression and regulation patterns (transcriptional activation or repression); (4) we identified a novel alternatively spliced form of *MITF* in wild-type *Andrias davidianus* that is not present in other vertebrates; (5) a critical mutation on the *SLC24A5* gene identified in albino *Andrias davidianus*, which has not been reported in other animals, could directly affect pigmentation. With these findings in mind, as one of the few studies exploring the regulation of body color in *Andrias davidianus*, we investigated the molecular mechanisms underlying the regulation of body color in *Andrias davidianus* and, in particular, found that the regulation of body color in *Andrias davidianus* differed from those in other vertebrates, which will certainly advance the application of molecular biology in the propagation and breeding of *Andrias davidianus* with specific body color and the ability to stably inherit to enrich the aquaculture market.

## 2. Materials and Methods

### 2.1. Experimental Animals

Wild-type and albino *Andrias davidianus* (two individuals, respectively) were purchased from a *Andrias davidianus* breeding base in Luoyang, Henan Province. All animal experiments were conducted in accordance with the guidelines and approval of the Animal Care and Welfare Committee of Luoyang Normal University (Approval code is 0020080A).

### 2.2. mRNA and sRNA Sequencing

The degradation and contamination of RNA, particularly DNA contamination, were monitored using 1.5% agarose gels. The NanoDrop 2000 Spectrophotometer (Thermo Fisher Scientific, Wilmington, DE, USA) was employed to determine the concentration and purity of RNA. The integrity of RNA was evaluated using the RNA Nano 6000 Assay Kit on the Agilent Bioanalyzer 2100 System (Agilent Technologies, Santa Clara, CA, USA).

For mRNA sequencing, a total amount of 1.5 μg of RNA per sample was used as input material for rRNA removal using the Ribo-Zero rRNA Removal Kit (Epicentre, Madison, WI, USA). The NEBNextR UltraTM Directional RNA Library Prep Kit for IlluminaR (NEB, Ipswich, MA, USA) was utilized according to the manufacturer’s instructions to produce sequencing libraries, with index codes added to assign sequences to individual samples. The protocol involved fragmenting the RNA with divalent cations at high temperature in the NEBNext First-Strand Synthesis Reaction Buffer (5×), followed by synthesis of the first-strand cDNA using random hexamer primers and Reverse Transcriptase. Subsequently, DNA Polymerase I and RNase H were used for second-strand cDNA synthesis, and exonuclease/polymerase activities were employed to convert any remaining overhangs into blunt ends. Adenylation of the 3′ ends of DNA fragments was performed, and NEBNext Adaptors with hairpin loop structures were ligated in preparation for hybridization. To ensure the selection of insert fragments ranging from 150 to 200 bp in length, the library fragments underwent purification using AMPure XP Beads from Beckman Coulter, located in Beverly, MA, USA. The cDNA that had been size-selected and adaptor-ligated was treated with 3 μL of USER Enzyme (NEB, Ipswich, MA, USA) at 37 °C for 15 min prior to PCR. The PCR was carried out using Phusion High-Fidelity DNA polymerase, along with Universal PCR primers and Index Primer. Lastly, PCR products were purified (AMPure XP system) and library quality was assessed on the Agilent Bioanalyzer 2100 and qPCR. 

To conduct sRNA sequencing, 2.5 ng of RNA was utilized as input material for RNA sample preparations. The NEBNextR UltraTM small RNA Sample Library Prep Kit for IlluminaR (NEB, Ipswich, MA, USA) was employed to generate sequencing libraries, following the manufacturer’s guidelines. Index codes were assigned to associate sequences with each sample. The process involved ligating the 3′ SR Adaptor as the first step. To prepare for ligation of the 3′ SR Adaptor for Illumina sequencing, a mixture of the adaptor, RNA, and nuclease-free water was incubated at 70 °C for 2 min in a preheated thermal cycler. The tube containing the mixture was transferred to ice, and 3′ Ligation Reaction Buffer (2×) and 3′ Ligation Enzyme Mix were added to facilitate the ligation of the 3′ SR Adaptor. The reaction was incubated at 25 °C for 1 h in a thermal cycler. In order to prevent the formation of adaptor-dimers, the excess of 3′ SR Adaptor (that remains free after the 3′ ligation reaction) was hybridized with the SR RT Primer, which transformed the single-stranded DNA adaptor into a double-stranded DNA molecule. Ligation-mediated processes do not support dsDNAs as substrates. The second step involves ligating the 5′ SR Adaptor, followed by the synthesis of the first chain through reverse transcription. Subsequently, PCR amplification and size selection were carried out. For fragment screening purposes, PAGE gel electrophoresis was performed, and small RNA libraries were created by recycling rubber-cutting pieces. Finally, the PCR products were purified using the AMPure XP system, and the quality of the library was evaluated on the Agilent Bioanalyzer 2100 system.

### 2.3. De Novo Assembly and Sequence Annotation

The clean reads were assembled using the Trinity software (https://sourceforge.net/projects/trinityrnaseq/files/ (accessed on 1 July 2021), trinityrnaseq-2.1.1). As there is no reference genome, in order to identify genes with low expression or genes with only partial fragments that are measured, the combined assembly of all samples of the same species can make the assembly results more comprehensive and facilitate subsequent analyses. The transcript reads were first broken into kmer (kmer is a continuous DNA sequence; assuming a read length L and kmer size K, then the Kmer number for each read is L − K + 1). The high-frequency kmer is then used as the seed to build a contig based on overlap relationships and cluster contigs of alternative spliced or other parallel genes. A separate de Bruijn graph was constructed for each contigs set and then the reads were aligned back to remove the sequencing error path and use the dynamic programming scoring algorithm to obtain the final transcripts, while selecting the longest transcripts in each locus as UniGene for subsequent analysis.

### 2.4. Analysis of Differentially Expressed Genes

The alignment results were calculated with eXpress software (https://pachterlab.github.io/eXpress/, express-1.5.1-linux_x86_64 (accessed on 10 July 2021)) to obtain the read count number of each sample to each UniGene, convert them to FPKM (Fragments Per Kilobase Million), and then analyze the expression level of genes. Differential expression analysis was performed based on the UniGene expression abundance values of 2 samples, and the main analysis software was DESeq2 (https://bioconductor.org/packages/release/bioc/html/DESeq2.html (accessed on 20 July 2021), version 1.14.0). Genes with |logFoldChange| ≥1 and adjusted *p*-value (*p*_adj_) ≤ 0.05 were considered as the differentially expressed genes for further analysis.

### 2.5. Functional Annotation, Pathway Enrichment and Network Construction

To annotate gene function, the following databases were employed: Nr (NCBI non-redundant protein sequence database), Nt (NCBI nucleotide sequences database), SWISS-PROT (a non-redundant protein sequence database that is manually curated), GO (Gene Ontology database), COG (Clusters of Orthologous Groups of proteins database), KOG (Clusters of Protein homology database), and KEGG (Kyoto Encyclopedia of Genes and Genomes database).

The clusterProfiler R package was employed to carry out Gene Ontology (GO) Enrichment Analysis of differentially expressed genes (DEGs), applying hypergeometric testing to identify significantly enriched GO entries in comparison to the entire genome background. 

The database KEGG (http://www.genome.jp/kegg/ (accessed on 21 July 2021)) was used for pathway analysis. The clusterProfiler R packages were utilized to identify KEGG pathways that were significantly enriched relative to the whole genome background. The DEG sequences were blasted (blastx) against the genome of a related species (which had protein–protein interaction data available on the STRING database, http://string-db.org/) to obtain the predicted protein–protein interaction (PPI) of the DEGs. Finally, Cytoscape was utilized to display the PPI network of these DEGs.

### 2.6. Reverse Transcription-Quantitative Polymerase Chain Reaction (RT-qPCR)

To verify the mRNA expression levels of selected genes, a quantitative RT-PCR (qRT-PCR) test was conducted. Reverse transcription was carried out in a 20 µL reaction mixture, which included 2 µL of total RNA, 1 µL of 5 µM RT primer, 1 µL of 10 mM dNTP, 4 µL of 5X PrimeScript Buffer, and 200 units of PrimeScript RTase (TaKaRa, Kusatsu, Shiga, Japan). The reaction was incubated at 42 °C for one hour and terminated by heating at 85 °C for 5 min. qPCR was performed with SYBR Green PCR Master Mix (TaKaRa, Kusatsu, Shiga, Japan) and all reactions were performed in triplicate. *GAPDH* was employed as the internal reference control.

### 2.7. Dendrogram Construction of MITF

The dendrogram of the relationships among the *MITF* of *Andrias davidianus* and other vertebrates were generated by SHOOT.bio (https://www.shoot.bio/, accessed on 5 October 2022). All the sequences were downloaded, and the sequence similarity was then calculated using the MEGA (https://www.megasoftware.net/, version 11) program to generate a branching pattern.

### 2.8. Protein Model Construction

A bHLH-Zip model of the MITF protein was constructed using SWISS-MODEL (online program accessed at https://swissmodel.expasy.org/, accessed on 1 September 2022)

## 3. Results

### 3.1. Overview of mRNA and miRNA Sequencing Data

To investigate the molecular mechanism of skin color variation in *Andrias davidianus*, we performed mRNA and small RNA sequencing of skin tissues from the wild-type (WT) and albino (AL) *Andrias davidianus* (Figure 1). The analysis of the mRNA sequencing data showed that 24,946,613 and 24,281,191 total sequences were obtained from two biological replicates in the AL group, and 25,071,096 and 24,314,935 total sequences were obtained from the WT group. After combined assembly of all samples of each group, 239,881 and 209,055 transcripts with N50 values of 1269 and 1257 were obtained in the AL and WT groups, respectively (Table 1). The average GC content was 48% and47%, respectively, and the percentages of Q30 bases were more than 94% for all the samples, suggesting high sequencing quality. Four databases including the SWISS-PROT database, Nr database, Nt database, and GO and KEGG database were used for gene prediction. The results showed that 29,407, 41,840, 21,058, 23,157 and 17,989 transcripts had significant hits against SWISS-PROT, Nr, Nt, GO, and KEGG, respectively (Table 1). Cumulatively, a total of 46,221 unique genes were predicted, allowing at least one significant hit against at least one of the three databases (Table 1).

Four small RNA libraries were constructed by deep sequencing. As shown in Table 2, 11.46 million (M) and 11.43 M total reads were obtained in the AL group. Subsequently to removing the low-quality and adaptor sequences, a total of 11.35 and 11.16 M clean reads were ultimately obtained. All identical sequence reads were then classified as groups, and 1.75 and 1.67 M unique reads were obtained. In these unique reads, 560,498 and 475,608 reads were annotated, respectively. In the WT group, 11.15 M and 11.05 M total reads were obtained, corresponding to 10.76 M and 10.52 M clean reads, and 1.08 and 1.06 M unique reads, respectively. In these unique reads, 481,482 and 447,731 reads were annotated, respectively. The novel and known miRNAs from each group are shown in Table 2 and Appendix A.

### 3.2. Enrichment and Pathway Analyses of Differentially Expressed Genes (DEGs)

Through analyzing all unique genes, 5517 unique genes were found to be differentially expressed between the wild-type and albino *Andrias davidianus* with |logFC| ≥1 and *p*_adj_ value ≤0.05 set as the criteria (Figure 2A). Among these DEGs, 2606 genes were down-regulated and 2911 genes were up-regulated in the wild-type *Andrias davidianus* compared with those in albino *Andrias davidianus* (Figure 2A and Appendix A). Gene set enrichment analysis and pathway analysis were performed on all DEGs to analyze the functional and regulation differences underlying the phenotypic variations. According to the GO terms, the DEGs were classified into three major functional categories, including 4743 DEGs in biological process (BP), 2046 DEGs in cellular component (CC), and 1861 DEGs in molecular function (MF) categories (Appendix A). The majority of the GO terms related to pigmentation and melanogenesis including pigment cell differentiation, melanin biosynthetic and metabolic processes, developmental pigmentation, melanocyte differentiation, melanosome and pigment granules, and melanosome membranes were significantly enriched and encompassed most of the key genes regulating pigmentation (Table 3). However, among these genes, we did not detect the presence of transcription factors PAX3 and SOX10, which may affect pigmentation by regulating *MITF* in other vertebrates (these two transcription factors were poorly expressed in the skin of two types of *Andrias davidianus* with different body colors and there was no statistical difference between the two). Intriguingly, we found two SOX family transcription factors, SOX21 and SOX7, in the DEGs (Appendix A). As a transcription activator, *SOX7* was highly expressed in wild-type *Andrias davidianus* (2.46-fold change), while the transcription inhibitor *SOX21* was less expressed in wild-type *Andrias davidianus* (2.93-fold change), which was consistent with the high expression pattern of pigment regulatory genes in wild-type *Andrias davidianus*.

In the KEGG pathway analysis, the DEGs were involved in 279 pathways and the top 25 enriched pathways are shown in Figure 2B. In these significantly enriched pathways, we found two melanin-associated signaling pathways, tyrosine metabolism and the MAPK signaling pathway (Figure 2B, Table 4 and Appendix A). However, the Wnt signaling pathway, another important melanin-related pathway in other vertebrates, was not identified, and the majority of the Wnt-signaling-pathway-related genes were unchanged in *Andrias davidianus* with different body colors (Table 4). It was suggested that the Wnt signaling pathway might not exert the same critical role in *Andrias davidianus* as it does in other vertebrates. Interestingly, we found a spliceosome in the significantly enriched pathways (Figure 2B), suggesting a key role of gene splicing in regulating pigmentation in *Andrias davidianus*. We indeed found a novel splicing form of the *MITF* gene in wild-type *Andrias davidianus* that is absent in other animals (see Section 3.4). This unique splicing pattern might directly affect the skin pigmentation in *Andrias davidianus*. A regulatory network was constructed using the DEGs related to pigmentation in *Andrias davidianus*, in which all of the genes are divided into three parts: MAPK signaling pathway-related genes, melanin synthesis and metabolism-related genes, and transcription factors (Figure 2C). *MITF* is the ‘hub gene’ that plays a critical role in pigmentation of *Andrias davidianus*. To confirm the reliability of the RNA-seq, 10 DEGs were chosen for validation by RT-qPCR. The expression levels of all 10 DEGs determined by RT-qPCR were concordant with their mRNA sequencing data (Figure 2D), indicating a strong association between mRNA profiling and the RT-qPCR data.

### 3.3. Integrated Analysis of Differentially Expressed miRNAs and mRNAs

To identify miRNAs that may be involved in melanin synthesis, we comprehensively analyzed the differentially expressed miRNAs and mRNAs. A total of 25 differentially expressed miRNAs were identified based on the criteria of |logFC| ≥ 1 and *p*_adj_ value ≤ 0.05 (Figure 3). Among these miRNAs, 17 downregulated and 8 upregulated miRNAs were found in the wild-type *Andrias davidianus* compared to the albino *Andrias davidianus* (Figure 3). We then obtained 5000 putative target genes of these differentially expressed miRNAs (Appendix A). As previously mentioned, there were 2606 downregulated genes and 2911 upregulated genes in the wild-type *Andrias davidianus* relative to the albino *Andrias davidianus*. Then, we intersected the target genes of 17 downregulated miRNAs with 2911 upregulated genes, yielding 99 intersection genes (Appendix A). Next, the target genes of 8 upregulated miRNAs were intersected with 2606 downregulated genes, from which 55 intersection genes were acquired (Appendix A). Subsequent analysis of these intersection genes showed the absence of the key genes that could regulate melanin synthesis yet the presence of only a few MAPK signal pathway-related genes. These findings suggest that miRNAs do not play a key role in the regulation of body color in *Andrias davidianus*.

### 3.4. Splicing of MITF and Mutation of SLC24A5

As mentioned above, we found the spliceosome to be the significantly enriched pathway (Figure 2B). To screen the genes affected by DEGs related to the spliceosome, we scanned the sequences of all mRNAs that might be involved in the regulation of pigmentation in *Andrias davidianus*. The results showed a unique splicing form of the *MITF* gene in the wild-type *Andrias davidianus*, which has not been reported in other animals (Figure 4, Figure 5 and Figure 6). More precisely, through the sequence alignment of *MITF* among 328 different animals (Appendix A), we identified deletion in the 3rd and 5th exons of *MITF* in *Andrias davidianus* (either wild-type or albino, Figure 4). More importantly, compared with the albino *Andrias davidianus*, a form of *MITF* mRNA in the wild-type *Andrias davidianus* was characterized as the insertion of 90 nucleotides between the 7th and 8th exons (30 extra amino acid residues of the protein) (Figure 4 and Figure 5A). The phylogenetic tree depicted the distance between the *MITF* in *Andrias davidianus* and their orthologs from vertebrates, which indicated that the *MITF* in *Andrias davidianus* belonged to a member of the MIiF-TEF family and belonged to the same branch of the evolutionary tree as the *MITF* protein of *Xenopus laevis* (Figure 5B). Interestingly, the *MITF* in *Andrias davidianus* was more closely related to that in terrestrial animals than in aquatic animals (Figure 5B). By analyzing this uniquely spliced form of the *MITF* protein in wild-type *Andrias davidianus*, we found 30 extra amino acid residues located in the loop of the basic/helix-loop-helix/leucine zipper (bHLH-Zip) structure of the *MITF* protein (Figure 5A). bHLH-Zip transcription factors usually function as dimers. A bHLH-Zip model of the *MITF* protein was constructed using SWISS-MODEL (Figure 5C), which revealed a longer loop structure for the wild-type *MITF* protein that might allow this protein to form a dimer with another helix through more flexible folding and packaging, or the DNA-binding domain of the dimer to have a more flexible and open spatial structure, thereby exerting regulatory effects on its downstream target genes and affecting the pigmentation process in *Andrias davidianus*.

After scanning all mRNA sequences that might be involved in the regulation of pigmentation in *Andrias davidianus*, we identified an interesting mutation in *SLC24A5* in the skin of albino *Andrias davidianus* (Figure 7). The mutation was presented in a cysteine-rich region (CCTCC) on the intracellular loop of the transmembrane protein NCKX5 encoded by *SLC24A5*, named the 4C region (Figure 7). Similar cysteine-rich regions also exist in other animals (Figure 7). This mutation was manifested as a loss of three amino acid residues of CCT in the 4C region in the albino *Andrias davidianus* (Figure 7). Cysteine residues play a crucial role in many proteins, particularly in enzyme reactions and intermolecular/intramolecular interactions [22]. In *Xenopus laevis*, the 4C mutant in NCKX5 could not rescue the reduction in pigmentation caused by NCKX5 knockdown [23], indicating a critical role of 4C in the functionality of NCKX5. The mutation in the 4C region of the NCKX5 protein in *Andrias davidianus* might be another major cause of skin albinism in addition to the alternative splicing of *MITF*.

## 4. Discussion

### 4.1. Key Genes Regulating Pigmentation in Andrias davidianus

Melanin is an amino-acid-derived biological pigment synthesized from melanocytes and is a polyphenolic polymer that can be mainly classified into two types: brown/black eumelanin and red/yellow pheomelanin [24]. Existing studies have shown that the pigmentation of melanin consists of four stages: melanosome formation, melanosome maturation, melanin synthesis, and final transfer and deposition of melanosomes containing a great deal of melanin in the skin keratinocytes, through which the skin color is presented [25]. Therefore, melanin pigmentation in the skin is closely related to melanosome formation, melanin synthesis, melanosome transportation, and the transcription activation of related genes at various stages. In this study, we conducted GO analysis on the DEGs between wild-type and albino *Andrias davidianus* and found that these DEGs involved multiple processes related to pigmentation, including pigment cell differentiation, melanin biosynthetic and metabolic processes, developmental pigmentation, melanocyte differentiation, melanosome and pigment granules, and melanosome membranes. After detailed analysis, these DEGs were demonstrated to participate in almost all of the processes of melanin pigmentation consisting of four stages (Figure 8). (1) Melanosome formation. Melanosome is a specialized membrane-bound organelle with striatal structures formed by amyloid fibers in melanocytes [26]. A pigment-cell-specific protein, the premelanosome protein (PMEL)—also known as SILV—plays a key role in the formation of melanosomes [27,28]. SILV is one of the molecules essential for the formation of melanosome fibers, which can independently assemble and form the striatal structure of melanosomes and maintain the environmental balance in melanosomes [27]. Studies have illustrated that SILV gathers on the intraluminal vesicles (ILVs) of melanosomes and gradually forms amyloid fibers with the elongation of ILVs [28,29]. A large number of amyloid fibers are packed into sheets and eventually form oval melanosomes [29]. (2) Melanosome maturation. The ion-exchange proteins encoded by the *OCA2*, *SLC24A5*, and *SLC45A2* genes play a key role in maintaining the internal environment stability of the melanosomes during melanosome maturation. They can regulate the concentration of positive ions such as Na^+^, K^+^, and H^+^ in melanosomes and jointly maintain the acid–base balance in the environment of melanosomes [13,30,31]. Their absence causes an abnormal morphology of melanosomes and amyloid fiber formation disorder, leading to a reduction in the melanin content [32]. In addition, RAB38 can control the transfer of melanin synthase (TYR, DCT, and TYRP1) to melanosomes, thus regulating the maturation of melanosomes [33,34]. (3) Melanin synthesis. The synthesis of melanin, especially the eumelanin, which determines the animal skin color (black or brown), is a continuous enzymatic reaction process. Melanin production is initiated from the TYR-catalyzed synthesis of dopa-quinone from L-tyrosine. Specifically, tyrosine is catalyzed by TYR and transformed into dopa (i.e., L-3,4-dihydroxyphenylalanine, L-DOPA), and L-DOPA is further oxidized to L-dopaquinone in the presence of TYR. In the absence of cysteine, L-dopaquinone undergoes cyclization to generate dopachrome, which in turn forms the intermediates 5,6-dihydroxyindole (DHI) and 5,6-dihydroxyindole-2-carboxylic acid (DHICA) through carboxylation and decarboxylation under the action of DCT. Then, DHI and DHICA form the two intermediates 5,6-indolequinone (IQ) and indole-5,6-quinone-2-carboxylic acid (IQCA) under the action of TYR and TYRP1. Finally, the heteropolymer formed through the binding of IQ to IQCA is regarded as eumelanin [4,35]. During this process, TYR, DCT, and TYRP1 are key rate-limiting enzymes and the expression of all three enzymes is regulated by the transcription factor *MITF* [32]. 4) Melanosome transportation. The transfer of melanosomes from melanocytes to keratinocytes is a key process to maintain skin pigmentation [36]. Rab GTPases play a role in the transfer of melanosomes, in which *RAB11A* knockout significantly affects the transfer of melanosomes [36,37]. In summary, the difference in skin color between wild-type and albino *Andrias davidianus* correlates with the gene expression changes during the whole formation to the transportation process of melanosomes. However, the differential expression of genes involved in the whole process does not seem to be regulated by miRNA. In fact, when we used these DEGs to construct a regulatory network, we found that *MITF* was a hub gene (Figure 2C), suggesting that *MITF* was one of the key genes for the skin color difference between two different types of *Andrias davidianus*.

### 4.2. Signaling Pathway Regulating Pigmentation in Andrias davidianus

Studies have illustrated that in most vertebrates, the Wnt/β-catenin and MAPK signaling pathways regulate pigmentation by modulating the transcription factor *MITF* [8,38]. In the MAPK signaling pathway, activated downstream ERK signaling molecules can inhibit *MITF* phosphorylation in an indirect or direct manner, thereby blocking the binding of *MITF* to downstream targets; moreover, the phosphorylation of MAPKs/ERK can increase the degradation of MITF, block the binding site (*MITF* M-box) on the tyrosinase promoter, and effectively inhibit the reduction in melanin synthesis caused by *MITF* transcription [38,39]. In the Wnt/β-catenin signaling pathway, accumulated β-catenin in melanocytes enters the nucleus and binds to LEF1, thereby acting on the promoter region of *MITF* and affecting its transcription level [8]. However, in this study, the analysis of the above two signaling-related genes found that the vast majority of the Wnt-signaling-related genes did not change in the skin of the wild-type and albino *Andrias davidianus* (Table 4), which suggests that the Wnt signaling pathway does not play a key role in the process of skin pigmentation in *Andrias davidianus* as it does in other vertebrates.

### 4.3. SOX Family and Pigmentation Regulation

The SOX family includes approximately 20 transcription factors, which contain a high-mobility-group (HMG) domain that binds to downstream DNA sequences [40]. In mammals, members of the SOX family can be subclassified into nine groups, while the HMG domain is highly conservative in the SOX family, with a ≥90% identity in the amino acid sequences of SOX proteins in the same group but approximately a 60% identity in the amino acid sequences of SOX proteins among different groups [41]. In vitro experiments have demonstrated that the HMG domains of SRY, SOX5, SOX9, and SOX17 have the same core binding sequence (5′-AACAAT-3′) [42,43,44,45]. In melanocytes, *MITF*, *DCT*, and *TYR* are common target genes of SOX10 and SOX9 [46], while SOX5, another member of the SOX family, can competitively bind to SOX10 target genes such as *MITF* and *DCT*, thus restraining the melanin synthesis [47]. These results indicate that SOX family members have overlapping target genes, and the role of several SOX family members may be substantiated or suppressed by other members.

Studies have unveiled that SOX10 plays a crucial role in melanin synthesis in the majority of vertebrates. Its role is manifested in two aspects: (1) SOX10 and PAX3 co-activate the expression of *MITF* [48], which affects the synthesis of melanin by regulating the expression of *TYR*, *DCT*, *TYRP1*, etc. [10,48,49]. (2) SOX10 can bind to the regulatory genes of melanin synthesis, such as *TYR* [50], *DCT* [51], and *TYRP1* [52], synergistically with *MITF* to activate the expression of these genes [51], while *MITF* cannot induce their expression individually [50,53]. However, the role of SOX10 in melanin synthesis is also challenged. It is believed that once the expression of *MITF* is established in melanocytes, other members of the SOX family substitute for the role of SOX10 in later development [54]. It has been demonstrated that after *SOX10* mutation in zebrafish, *MITF* can induce the expression of *TYR* independent of SOX10 and completely rescue pigmentation [50]. In this study, we found that the expression of SOX10 and its partner transcription factor PAX3 was very low in the skin of both the wild-type and albino *Andrias davidianus* (especially PAX3, which was almost not expressed) and exhibited no difference between the two different types of *Andrias davidianus*. In melanocytes, SOX10 and PAX3 form a complex to achieve the transcriptional regulation function, while SOX10 alone cannot activate the expression of downstream genes such as *MITF* [48], indicating that SOX10 and its partner transcription factor PAX3 have an extremely limited role in the pigmentation process of the skin of *Andrias davidianus*. Interestingly, we analyzed other members of the SOX family and found that the other two members, SOX7 and SOX21, were highly expressed in the skin of wild-type and albino *Andrias davidianus* and were differentially expressed between the two. SOX7, belonging to group F of the SOX family, usually contributes to the transcriptional activation of target genes, while SOX21, belonging to group B, acts as a transcriptional inhibitor [55]. The expression level of *SOX7* was 2.46-fold upregulated in WT skin compared with AL, while the *SOX21* expression in the AL skin was 2.93-fold upregulated compared with WT (Appendix A), suggesting that SOX7 and SOX21 could act as transcriptional activators and inhibitors in the pigmentation of the skin of *Andrias davidianus*. Whether SOX7 acts as a substitute for SOX10 to activate the expression of *MITF* and genes related to melanin synthesis, such as *TRY*, *DCT*, and *TYRP1*, or whether SOX21 acts as an inhibitor to inhibit the expression of these genes remains to be further proved.

### 4.4. MITF Splicing

As a transcriptional activator, *MITF* can regulate the expression of multiple genes related to pigmentation, including melanosome assembly and melanin synthesis [32]. *MITF* consists of at least four isomers, MITF-A, MITF-C, MITF-H, and MITF-M [56,57], which show differences in the amino-terminal but share the entire carboxyl portion encoded by exons 2 to 9 of the *MITF* gene [58]. Among them, MITF-M is only expressed in melanocytes and melanoma cells, which is a lineage-specific isoform in melanocytes [56,57]. There are two major isoforms of MITF-M with and without six amino acids (6a, ACIFPT) inserted in exon 6, named (+) *MITF* and (−) *MITF*, respectively [56,57,59]. In this study, we found that the *MITF* in *Andrias davidianus* showed great uniqueness compared with other vertebrates. By comparing the *MITF* protein sequences of *Andrias davidianus* and 328 other species, we found its uniqueness mainly presented in the following three aspects: (1) Exon 3 and exon 5 of *MITF* were deleted simultaneously in *Andrias davidianus* (Figure 4 and Appendix A). Compared with common *MITF* protein sequences, the deletion of exon 3 also exists in amphibians such as Xenopus tropicalis, *Protopterus annectens*, *Bufo bufo*, *Nanorana parkeri,* and *Rana temporaria* (Figure 6A). However, exon 5 deletion was not presented, particularly the deletion of both exon 3 and exon 5 in *Andrias davidianus*. (2) *MITF* in most animals either contains 6a or does not contain 6a, while the sequence at 6a of *MITF* was altered to DATVYY in *Andrias davidianus* (Figure 6B). Sequence changes at 6a of *MITF* also exist in other animals such as *Microcaecilia unicolor*, *Bufo gargarizans*, *Geotrypetes seraphini,* and *Amblyraja radiata*, but these sequences have no consistency with each other (Figure 6B). (3) An insertion of 30 amino acids between exons 7 and 8 of *MITF* existed in the wild-type *Andrias davidianus* but not in albino *Andrias davidianus* (Figure 5A and Figure 6C). Insertions between exons 7 and 8 are rare in other animals, and the insertion sequences found only in *Rana temporaria*, *Myotis lucifugus*, *Pogona vitticeps,* and *Apteryx rowi* are completely different from those in the wild-type *Andrias davidianus* (Figure 6C). Therefore, it could be concluded that the splicing process of the *MITF* in *Andrias davidianus* is more complicated than those in other vertebrates. Interestingly, through the analysis of the DEGs in wild-type and albino *Andrias davidianus*, we found that these DEGs were enriched in the spliceosome pathway (Figure 2B). These spliceosome-related DEGs might be the main reason why *MITF* showed different splicing forms in two types of *Andrias davidianus* with different skin colors. As previously mentioned, the insertion of 30 amino acids changed the bHLH-Zip structure (Figure 5A,C), thereby affecting the expression of the *MITF* downstream genes, which might be one of the major reasons for the differential expression of key melanin-related genes (such as *TYR*, *DCT*, *TYRP1*) in the skin of wild-type and albino *Andrias davidianus*.

### 4.5. SLC24A5 Mutation

The *SLC24A5* gene, the fifth member of the solute carrier family 24, encodes a potassium-ion-dependent cation exchange protein (NCKX5) located on the melanosome membrane. NCKX5 can transport calcium ions into the melanosome and pumps sodium ions out of the organelle, which plays a crucial role in maintaining the concentration gradient of hydrogen ions and calcium ions inside and outside the melanosome [60]. The formation of an ion gradient is involved in the synthesis of melanin in the melanosome [60]. The mutation of the *SLC24A5* gene regulates the PH in melanocytes and affects the maturation and catalytic activity of tyrosinase, resulting in different color phenotypes [13]. Additionally, intracellular calcium can affect the production of melanin by activating the PMEL protein [61].

The transmembrane protein NCKX5 contains a cytoplasmic loop, which divides the transmembrane structure of the NCKX5 protein into two transmembrane segments, TMS1 and TMS2 [62]. By comparing the cytoplasmic loop sequences of 470 different vertebrates (Appendix A), we found a cysteine-rich region (4C or 3C) that was conserved in all animals (Figure 7). Cysteine residues play an important role in many proteins, especially in enzyme reactions and intermolecular/intramolecular interactions [22]. The 4C or 3C region in the cytoplasmic loop of NCKX5 is believed to participate in the transition metal binding and S-acylation, thus affecting the localization of the NCKX5 protein on the membrane [63,64]. It was also indicated in another study that the mutation of a specific cysteine resulted in a significant reduction in the expression of NCKX2, another member of the NCKX family [65]. It was found in *Xenopus laevis* that the 4C mutant in NCKX5 could not rescue the reduction in pigmentation triggered by NCKX5 knockdown, indicating that 4C exerted a very important role in the function of NCKX5 [23]. In this study, we revealed that the mutation in the 4C region (CCTCC) of the albino *Andrias davidianus* caused the deletion of three amino acid residues of CCT (Figure 7). This mutation might be another major cause of skin albinism in addition to the alternative splicing of *MITF*.

## 5. Conclusions

In this study, through mRNA and small RNA sequencing analyses of wild-type and albino *Andrias davidianus*, we identified the genes and signaling pathways involved in melanin synthesis, pigmentation, etc., and found that *Andrias davidianus* has distinct regulatory mechanisms that differed from other vertebrates as follows. (1) The difference in body color between the wild-type and albino *Andrias davidianus* did not seem to be significantly influenced by miRNAs. (2) The Wnt signaling pathway, the key signaling pathway related to melanin synthesis in other vertebrates, does not play a significant role in *Andrias davidianus* (Figure 7). (3) *MITF* acts as the core transcription factor that influences body color in *Andrias davidianus*; however, its regulator, SOX10, as well as PAX3—the partner transcription factor of SOX10—do not play a role in the pigmentation process in *Andrias davidianus*. The expression of two other members of the SOX family, SOX7 and SOX21, significantly differs between the two types of *Andrias davidianus* (Figure 7). (4) *MITF* in *Andrias davidianus* has a unique splicing form, and specifically, the insertion of 30 amino acids in the bHLH-Zip structure might directly affect the functionality of MITF. (5) A unique mutation exists in the *SLC24A5* gene in *Andrias davidianus*, resulting in the deletion of the 4C region of the protein, thereby affecting the protein activity. The identification of the molecular mechanisms underlying the regulation of body color in *Andrias davidianus* will advance the utilization of molecular biology in the breeding of *Andrias davidianus* species with a specific body color that can be stably inherited to enrich the aquaculture market.

## Figures and Tables

**Figure 1 animals-13-01181-f001:**
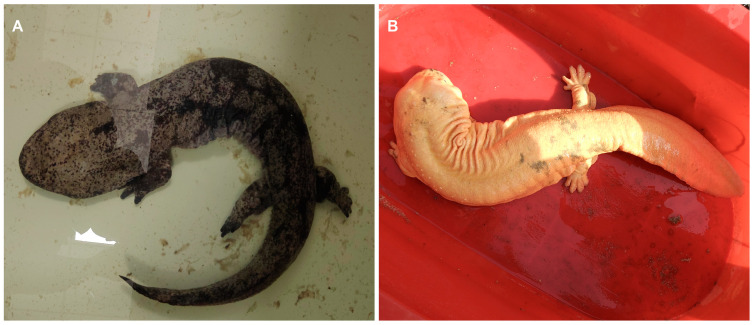
Two skin color types of *Andrias davidianus*. (**A**). Wild-type (WT) *Andrias davidianus*. (**B**). Albino (AL)-type *Andrias davidianus*.

**Figure 2 animals-13-01181-f002:**
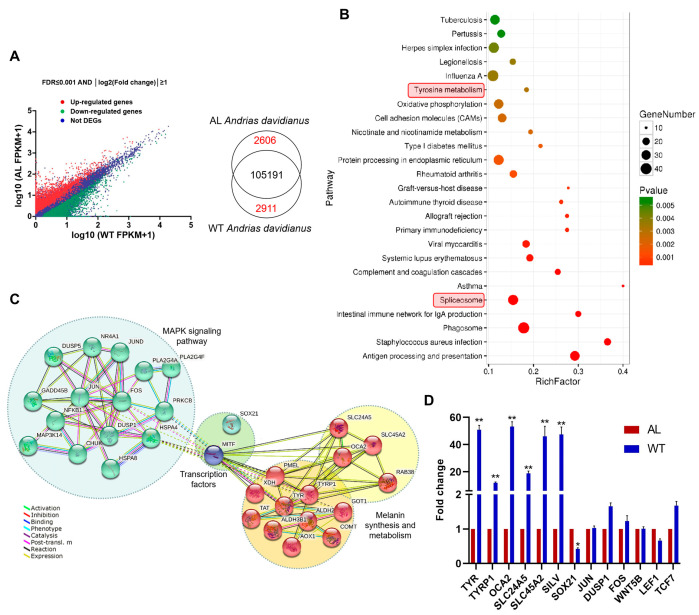
Sequencing analysis between wild-type (WT) and albino (AL) *Andrias davidianus* skin. (**A**) Genes differentially expressed in WT compared with AL *Andrias davidianus* skin. (**B**) KEGG enrichment analysis of DEGs shows the top 25 KEGG pathways. The red box marks the pathways associated with pigmentation and gene splicing. (**C**) Protein-to-protein interaction (PPI) network constructed by DEGs of pigmentation, melanogenesis, and pigmentation-related pathways. (**D**) Validation of selected genes using qRT-PCR. Fold changes are calculated by normalizing expression relative to *GAPDH*. * represents *p* value ≤ 0.05 and ** represents *p* value ≤ 0.01.

**Figure 3 animals-13-01181-f003:**
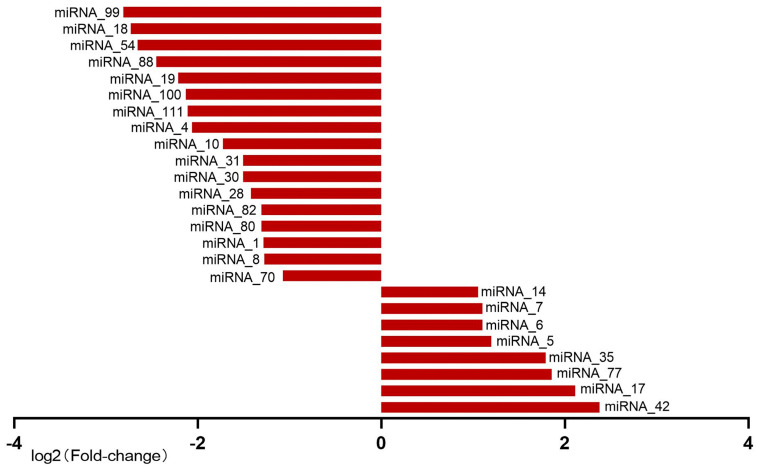
Differentially expressed miRNAs in WT compared with AL *Andrias davidianus* skin. A total of 25 differentially expressed miRNAs were identified based on the criteria of |logFC| ≥ 1 and *p*_adj_ value ≤ 0.05.

**Figure 4 animals-13-01181-f004:**
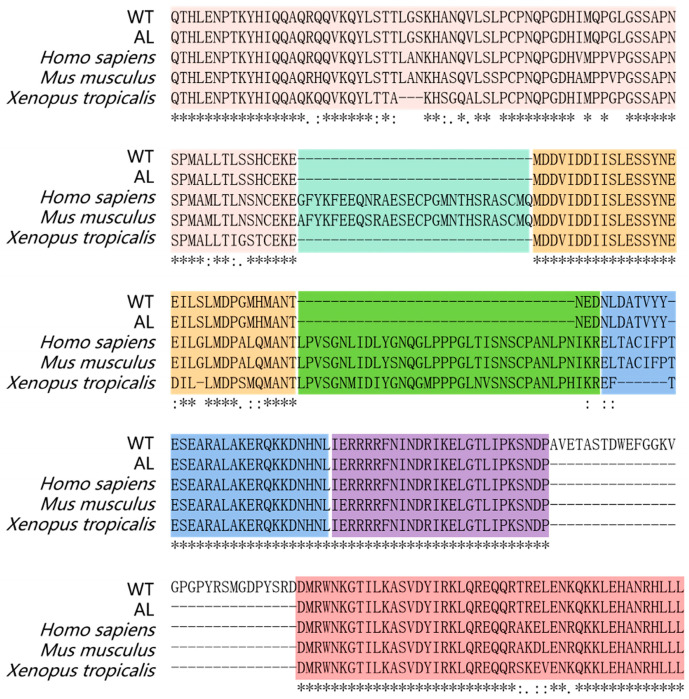
Protein sequence alignment of *MITF* between WT, AL *Andrias davidianus,* and other vertebrates. The colored boxes mark exons 2–8 of MITF.

**Figure 5 animals-13-01181-f005:**
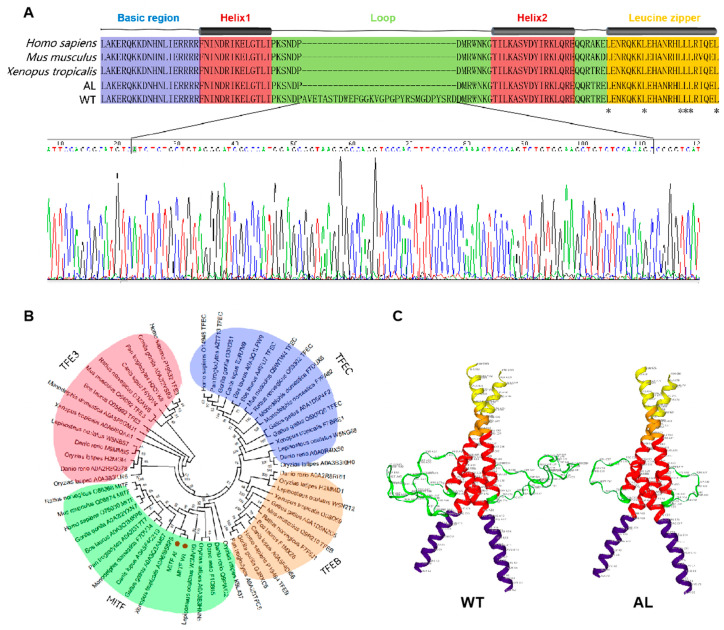
Differential splicing form of *MITF* between WT and AL *Andrias davidianus*. (**A**) Amino acid comparison of the bHLH-Zip domain of MITF, *Homo sapiens*, *Mus musculus*, *Xenopus tropicalis,* and WT and AL *Andrias davidianus*. (**B**) Dendrogram of the relationships among the *MITF* of *Andrias davidianus* from other vertebrates. The sequence similarity was calculated using the MEGA program to generate a branching pattern. The numbers below the branches indicate the percentages of bootstrap support after 1000 replicates. (**C**) Three-dimensional structure of the *MITF* bHLH-Zip domain (WT and AL) constructed by SWISS-MODEL.

**Figure 6 animals-13-01181-f006:**
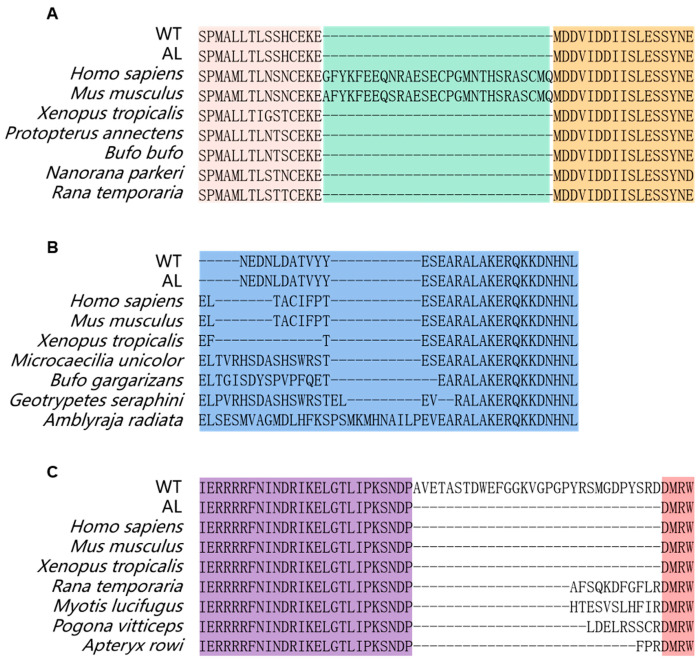
Unique structure of *MITF* in *Andrias davidianus* compared with other vertebrates. When comparing *MITF* protein sequences of *Andrias davidianus* with 328 other species, it was found that (**A**) Exon 3 of *MITF* was deleted in *Andrias davidianus* and a few other animals. (**B**) *MITF* in most animals either contains 6a (TACIFP) or does not contain 6a, while the sequence at 6a of *MITF* was altered to another sequence in the *Andrias davidianus* and a few other animals. (**C**) An insertion of 30 amino acids between exons 7 and 8 of *MITF* existed in the WT *Andrias davidianus* but not in albino *Andrias davidianus* and most other vertebrates.

**Figure 7 animals-13-01181-f007:**
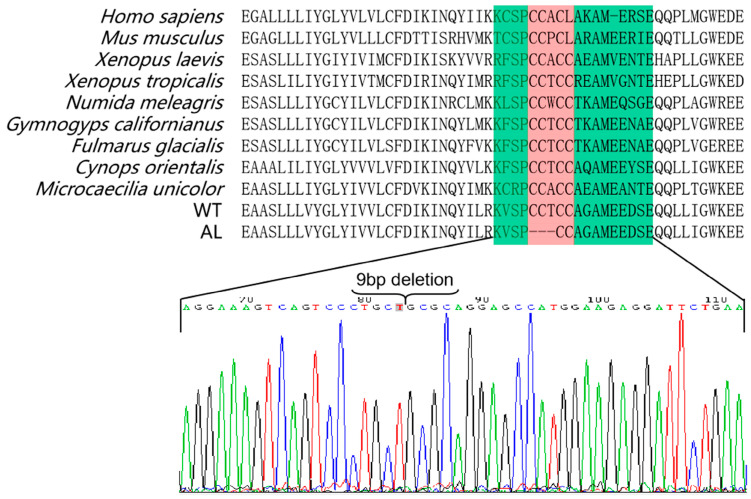
Protein sequence alignment of SLC24A5 between WT, AL *Andrias davidianus,* and other vertebrates. The red box marks the 3-amino-acids mutation of AL *Andrias davidianus*.

**Figure 8 animals-13-01181-f008:**
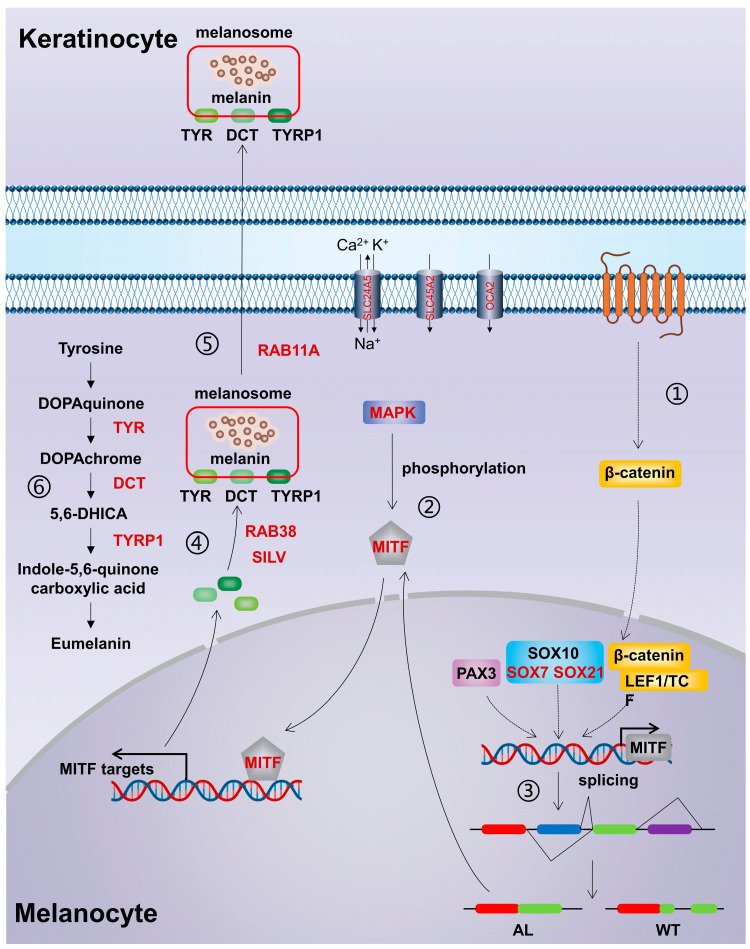
Diagram of putative gene pathways in the *Andrias davidianus* skin pigmentation process. ① Wnt signaling pathway; ② MAPK signaling pathway; ③ *MITF* splicing; ④ The formation and maturation process of melanosome; ⑤ Melanosome transportation; ⑥ The melanin biosynthesis pathway. The red marked genes represent DEGs in WT compared to AL *Andrias davidianus*. Dotted lines represent the events that are different or undefined in the pigmentation of *Andrias davidianus* compared with other vertebrates.

**Table 1 animals-13-01181-t001:** Overall statistics based on the reads of the mRNA sequencing data.

		AL-1	AL-2	WT-1	WT-2
Assembly	Total sequences	24,946,613	24,281,191	25,071,096	24,314,935
	Total number of transcripts	239,881		209,055	
	Contig N50 (bp)	1269		1257	
	Q30 (%)	95.6		94.9	
	GC (%)	48		47	
Annotation	No. of unique genes with Blast hit to Swissprot	29,407			
	No. of unique genes with Blast hit to Nr	41,840			
	No. of unique genes with Blast hit to Nt	21,058			
	No. of unique genes with GO term	23,157			
	No. of unique genes with KEGG	17,989			
	Total annotation unique genes	46,221			

**Table 2 animals-13-01181-t002:** Overall statistics based on the reads of the miRNA sequencing data.

	AL-1	Al-2	WT-1	WT-2
Raw reads	11,465,944	11,435,331	11,156,827	11,059,089
Clean reads	11,350,518	11,163,594	10,759,710	10,517,778
Unique reads	1,750,867	1,669,795	1,082,456	1,056,774
Mapped reads	4,277,994	4,929,921	5,787,977	6,015,248
Annotated reads	560,498	475,608	481,482	447,731
Known miRNAs	77	77	74	70
Novel miRNAs	32	32	32	31

**Table 3 animals-13-01181-t003:** DEGs and their GO terms related to pigmentation and melanogenesis in WT versus AL *Andrias davidianus* skin.

GO Terms	Genes	Relative Expression in WT vs. AL *Andrias davidianus* Skin	Fold Change (log2)	*p*-Value
Pigment cell differentiation	*SLC45A2*	Up-regulation	2.86	0.00026
	*TYRP1*	Up-regulation	4.45	6.55 × 10^−11^
	*RAB38*	Up-regulation	2.54	0.00131
	*OCA2*	Up-regulation	4.06	3.80 × 10^−8^
	*MITF*	Up-regulation	1.44	0.00141
	*TYR*	Up-regulation	4.08	3.09 × 10^−8^
	*SLC24A5*	Up-regulation	2.29	0.00268
Melanin biosynthetic and metabolic process	*TYRP1*	Up-regulation	4.45	6.55 × 10^−11^
	*OCA2*	Up-regulation	4.06	3.80 × 10^−8^
	*TYR*	Up-regulation	4.08	3.09 × 10^−8^
	*SLC24A5*	Up-regulation	2.29	0.00268
Developmental pigmentation	*TYRP1*	Up-regulation	4.45	6.55 × 10^−11^
	*SILV*	Up-regulation	3.31	1.44 × 10^−5^
	*RAB38*	Up-regulation	2.54	0.00131
	*OCA2*	Up-regulation	4.06	3.80 × 10^−8^
	*SPNS2*	Down-regulation	−1.57	0.00594
	*MITF*	Up-regulation	1.44	0.00141
	*TYR*	Up-regulation	4.08	3.09 × 10^−8^
	*SLC24A5*	Up-regulation	2.29	0.00268
	*SLC45A2*	Up-regulation	2.86	0.00268
Melanocyte differentiation	*TYRP1*	Up-regulation	4.45	6.55 × 10^−8^
	*MITF*	Up-regulation	1.44	0.00141
	*SLC24A5*	Up-regulation	2.29	0.00268
	*SLC45A2*	Up-regulation	2.86	0.00268
Melanosome and pigment granule	*P4HB*	Down-regulation	−7.47	2.83 × 10^−53^
	*TYR*	Up-regulation	4.08	3.09 × 10^−8^
	*YWHAZ*	Up-regulation	1.08	1.08 × 10^−7^
	*HSPA5*	Up-regulation	1.39	2.18 × 10^−15^
	*TYRP1*	Up-regulation	4.45	6.55 × 10^−11^
	*SILV*	Up-regulation	3.31	1.44 × 10^−5^
	*RAB38*	Up-regulation	2.54	0.00131
	*OCA2*	Up-regulation	4.06	3.80 × 10^−8^
	*HSP70*	Up-regulation	3.48	5.10 × 10^−6^
	*YWHAB*	Up-regulation	1.10	0.002411
	*RAB11A*	Up-regulation	1.60	3.84 × 10^−5^
	*CEP63*	Up-regulation	1.99	1.92 × 10^−19^
Melanosome membrane	*TYR*	Up-regulation	4.08	3.09 × 10^−8^
	*OCA2*	Up-regulation	4.06	3.80 × 10^−8^
	*TYRP1*	Up-regulation	4.45	6.55 × 10^−11^

**Table 4 animals-13-01181-t004:** Detailed information about the DEGs involved in each of the three pigmentation-related pathways.

GO Terms	Genes	Relative Expression in WT vs. AL *Andrias* *davidianus* Skin	Fold Change (log2)	*p*-Value
Tyrosine metabolism	*AOX1*	Down-regulation	−5.42	8.03 × 10^−16^
	*TYRP1*	Up-regulation	4.45	6.55 × 10^−11^
	*ALDH3B1*	Down-regulation	−2.95	0.00016
	*COMT*	Down-regulation	−1.82	3.32 × 10^−14^
	*XDH*	Down-regulation	−2.18	2.43 × 10^−5^
	*ALDH2*	Up-regulation	1.63	8.27 × 10^−12^
	*TAT*	Up-regulation	1.74	1.10 × 10^−10^
	*TYR*	Up-regulation	4.08	3.09 × 10^−8^
	*GOT1*	Up-regulation	1.11	1.59 × 10^−8^
MAPK signaling	*JUN*	Up-regulation	3.06	6.19 × 10^−66^
	*HSPA8*	Up-regulation	2.99	2.73 × 10^−16^
	*PRKCB*	Down-regulation	−1.17	0.0012
	*PLA2G4A*	Up-regulation	1.30	0.0015
	*CHUK*	Down-regulation	−1.02	2.01 × 10^−6^
	*PLA2G4F*	Down-regulation	−1.003	7.90 × 10^−6^
	*NR4A1*	Up-regulation	4.91	4.75 × 10^−114^
	*NFKB1*	Down-regulation	−1.02	0.0002
	*FOS*	Up-regulation	5.17	4.64 × 10^−68^
	*DUSP1*	Up-regulation	4.23	2.04 × 10^−101^
	*GADD45B*	Down-regulation	−3.08	7.26 × 10^−5^
	*MAP3K14*	Down-regulation	−1.92	0.0054
	*DUSP5*	Up-regulation	1.84	0.0016
	*HSP70*	Up-regulation	3.48	5.10 × 10^−6^
	*JUND*	Up-regulation	1.93	3.15 × 10^−27^
	*MAP4K5*	Down-regulation	−1.20	3.22 × 10^−9^
Wnt signaling	*WNT5B*	Unchanged	−0.82	0.1533
	*SFRP5*	Unchanged	2.15	0.1582
	*CAMK2*	Unchanged	0.22	0.8597
	*CTBP*	Unchanged	0.036	0.8304
	*DAAM*	Unchanged	−0.047	0.9311
	*EP300*	Unchanged	−0.17	0.6570
	*NKD*	Unchanged	0.21	0.2158
	*NLK*	Unchanged	0.029	0.9413
	*CCND1*	Unchanged	−0.34	0.3126
	*VANGL*	Unchanged	0.74	0.3622
	*LEF1*	Unchanged	−0.040	0.1024
	*TCF7*	Unchanged	−0.381	0.1638
	*FZD2*	Unchanged	0.541	0.1548
	*DVL1*	Unchanged	0.787	0.1087
	*GSK-3* *β*	Unchanged	−0.877	0.1568

## Data Availability

The data presented in this study are available in Appendix A.

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
