# Peer review of "Coordinated microRNA/mRNA Expression Profiles Reveal Unique Skin Color Regulatory Mechanisms in Chinese Giant Salamander (Andrias davidianus)"

_animals, 2023, doi:10.3390/ani13071181_

Round 1

Reviewer 1 Report

In this study, Guo et al. investigate the genetic origin of the albino morph in the Chinese giant salamander using transcriptome and small RNA analyses. They find genes and miRNAs differentially expressed between albino and wild type individuals. They also find an alternative splicing form of MITF and a mutation in SLC24A5 gene, and hypothesize that it might be the main cause of albinism in this species. 

Overall, the paper is well written and the topic is of interest to researchers studying pigmentation. I think that the paper would benefit from giving more context for the different analyses, with a clearer narrative on why each analysis was conducted. For example, it is not completely clear to me why the authors conducted an analysis to identify SSRs. Also, being on skin coloration, I think that photos of the two phenotypes should be shown, and if possible, histology/TEM of the skin to illustrate the difference in melanophore, melanosomes and melanine between the morphs.

Small language adjustments should be made throughout the manuscript. For example, I don't think that it is correct to use the terms "cultivars" and "cultivation" when referring to animals. 

Finally, please make sure that the text within the figures is readable. Some of it is extremely small. 

Reviewer 2 Report

The paper: "Coordinated microRNA/mRNA expression profiles reveals unique skin color regulatory mechanisms in Chinese giant salamander (Andrias davidianus)" describes an attempt to describe the mechanism by which the albino phenotype arises in the salamander. The issue is interesting and worth investigating. However, the presented work contains numerous methodological errors that reduce the credibility of the conclusions. There is a lack of information on number of samples used in RNA-seq, sRNA-seq, qPCR and Sanger sequencing. It seems that in RNA-seq only one sample of WT and one sample of Albino was used which is much too small. The deletion identification seems to be better supported, however there still lack of clear indication about the number of sample (supplementary Fig. 7 is not enough)

Morre specific comments:

line 10 distinct mechanism of what?

line 13 MITF gene?

line 62 please explain the shortcut

Please write the gene names in italics

line 90 please rewrite the sentence 

line 108 please provide the number of animals and the number of ethical permission

line 142 Please correct the tense

Materials and methods:

Please provide the description of bioinformatic analysis of NGS data

line 154 what kind of contigs? any accession numbers?

Is the direction of gene expression changes as expected ?

Round 2

Reviewer 2 Report

All my questions have been answered